# Cascaded Dilated Dense Network with Two-step Data Consistency for MRI Reconstruction

**Hao Zheng, Faming Fang,**[*] **Guixu Zhang**
Shanghai Key Laboratory of Multidimensional Information Processing,
and the School of Computer Science and Technology
East China Normal University
wsnbzh@hotmail.com, {fmfang, gxzhang}@cs.ecnu.edu.cn

## Abstract

Compressed Sensing MRI (CS-MRI) aims at reconstrcuting de-aliased images from sub-Nyquist sampling $k$-space data to accelerate MR Imaging. Inspired by recent deep learning methods, we propose a Cascaded Dilated Dense Network (CDDN) for MRI reconstruction. Dense blocks with residual connection are used to restore clear images step by step and dilated convolution is introduced for expanding receptive field without taking more network parameters. After each sub-network, we use a novel Two-step Data Consistency (TDC) operation in $k$-space. We convert the complex result from first DC operation to real-valued images and applied another replacement with sampled $k$-space data. Extensive experiments demonstrate that the proposed CDDN with TDC achieves state-of-art result.

## 1 Introduction

Magnetic resonance imaging (MRI) [11] is widely used in clinical diagnosis. It extracts internal information of the human body to detect latent lesion. Unlike conventional imaging techniques, MRI gathers phase-encoding data from $k$-space instead of image domain. The scanning procecss should follow the Nyquist criteria [14] to produce clear images, but it leads to long acquisition time. Patients can have tension as they have to keep still in the entire process.

Sub-Nyquist sampling can significantly reduce the acquisition time by skipping partial phase information, but it leads to aliased artifacts. In order to recovery clear image from sub-sampled $k$-space data, CS-MRI approaches were proposed [2]. With the assumption that MR images are sparsity in specific transfrom domain, classic sparsity-prior methods apply transfroms like discrete Fourier transform (DFT) [5], discrete cosine transfrom (DCT) [20, 25] and discrete wavelet transform (DWT) [10, 16]. Data-driven methods (i.e. dictionary learning) achieve higher accuracy due to the adaptive feature representation learnt from a quantity of fully sampled data [33]. Although these methods success in restoring clear image, they still suffer from heavy computation overhead.

Recent years, deep learning has achieved excellent result in a variety of image-restoring problem such as de-noising [34], de-blurring [28], super-resolution [26], etc. Generally, deep learning methods develop deep neural network to learn the mapping function from one distribution to another one. In MRI reconstruction, a common way is training a convolution neural network (CNN) for mapping from aliased images (directly reconstructed from zero-filled sub-sampled $k$-space data) to corresponding clear images [24]. U-Net is a popular framework in medical image processing [19].

It also accomplished accurate result on MRI reconstruction [7, 17, 29]. Yang *et al.* [30] proposed ADMM-CSNet to learn the parameters of ADMM algorithm with nerual network instead of manual adjustment. More recently, cascading network was introduced to MRI reconstruction [21, 22].

---

[*]Corresponding Author

Unlike normal image restoration, original data of MRI is acquired from $k$-space. Frequency domain data consistency plays an important role in MRI reconstruction. Hyun *et al.* [7] directly replaced the corresponding phasing-encoding data with sampled data. Yang *et al.* [29] used frequency domain loss while Quan *et al.* [17] applied cyclic loss. Schlemper *et al.* [21] implemented a consistency layer with a noise-adaptive parameter for noisy data. For cascading network, such data consistency layer provides intermedia information correction between sub-networks.

In this paper, we propose a novel network architecture called Cascaded Dilated Dense Network with Two-step Data Consistency layer (CDDNwithTDC). Our contributions can be summarized as follows:

(1) We use cascaded dense blocks to reconstruct MR images to improve performance as well as reduce the number of parameters. Such intra-block dense shortcut architecture alleviates the gradient vanishment and preserves detail information.

(2) We introduce dilated convolution to dense blocks, which expands receptive field without any additional parameters. The combination sufficiently extracts latent information.

(3) We propose a Two-step Data Consistency layer to enhance the naturalness of MR images while ensuring the data consistency in $k$-space.

Numerous experiments show the advancement of our proposed method in MRI reconstruction. [2]

## 2 Related Works

### 2.1 Cascaded Network

Cascaded network uses a serial of sub-networks to process data step by step. The later sub-networks take the former result as input to improve the accuracy. Quan *et al.* [17] proposed RefineGAN by cascading two U-Net as generator. As simply cascading network has no difference with naively increasing the depth of network, it can easily reach a bottleneck of performance. In MRI reconstruction, data consistency operation can be applied as a postprocess of sub-network, which replace the specific $k$-space position with the sampled value [22]. Such operation enable skip connection between input and each sub-network to alleviate gradient vanishment.

### 2.2 Dense Connection

In general, a deeper network has higher performance, but it also suffers from gradient vanish problem. After a long chain of gradient backward , the gradient information in the early stage can be too small for updating parameters. Skip connection alleviates such phenomenon as mentioned before. Dense connection applies shortcuts among all the layers [6]. In a dense block (a number of convolution layers with dense connection), the input of each layer is the concatenation of all the previous layers' output. Free data flow in dense block benefits the robustness of network. Tong *et al.* [23] applied dense connection by cascading dense blocks for image super-resolution task. Li *et al.* [12] implemented dense connection with U-Net. Although it brings additional network parameters, dense connection is a worthy trade-off. And in this paper, we will limit the network parameters (like reduce the number of intermedia feature channel) to show the superiority.

### 2.3 Dilated Convolution

Yu *et al.* [31] firstly proposed dilated convolution for senmatic segmentation. Receptive field has sensitive connection with network's abilty of latent global information extraction. In classic convolution, deeper layers involve a combination of receptive fields from former layers, while these receptive fields have large overlapping area. Dialted convolution applies hollow convolution kernel to alleviate overlapping with no more parameters. Moeskops *et al.* [13] introduced dilated convolution to brain MRI segmentation and proved that dilated network has larger receptive field with fewer network parameters than fully convolutional network. Perone *et al.* [15] applied parallel convolution with different dilation scales to abstract multi-scale information. Qiao *et al.* [35] proposed Pyramid Dilated Convolution Unit as a birdge to connect the encoder and the decoder of U-Net. Sun *et al.* [22] adopt dilated convolution in cascading blocks.

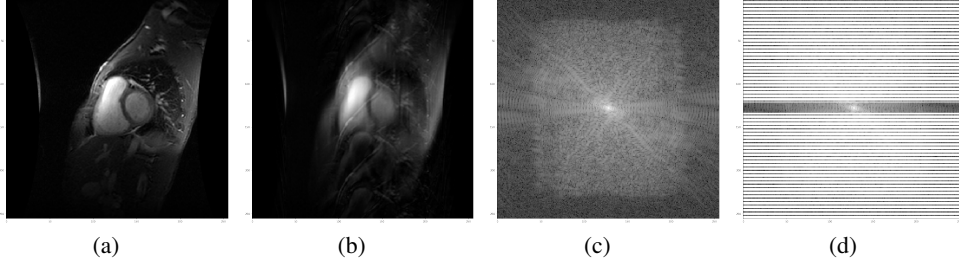

Figure 1: Sub-Nyquist sampled MR Image. (a) Fully sampled MR Image $x$. (b) Zero-filled reconstructed Image $x_u$ from sub-Nyquist sampled $k$-space data (d). (c) The $k$-space data of the (a). (d) Acquired $k$-space data $y$ with a 28.5% sampling rate Cartesian mask.

## 3  Method

### 3.1  Problem Formulation

The problem is to reconstruct fully-sampled image from sub-sampled $k$-space data. With sub-Nyquist sampling, the acquisition process can be written as:

$$y = M \odot Fx + \varepsilon \tag{1}$$

Here $x \in \mathbb{C}^{N_x \times N_y}$ is the original MR image (fully-sampled) to be reconstructed and $F$ is Fourier Transfrom operator. $M \in \mathbb{C}^{N_x \times N_y}$ is the sampling mask matrix composed of 1 and 0. The values of $M$ stand for the corresponding $k$-space positions are sampled or not. $\odot$ is pixel-wise multiply operation. Notice that sampling style is limited by MRI equipment. In this paper, we focus on the phase direction sampling, i.e. $M$ only contains 0-lines and 1-lines. $\varepsilon$ is the noise generated during acquisition and $y \in \mathbb{C}^{N_x \times N_y}$ is the $k$-space data what we actually observed. An example of sub-Nyquist sampled MR Image is given in Figure 1.

Unfortunately, Eq.1 is underdetermined. In order to solve the ill-posed inversion, conventional CS-MRI methods formulate an optimisation problem:

$$\hat{x} = \arg\min_x \|M \odot Fx - y\|_2^2 + \sum_i \lambda_i \psi_i(x) \tag{2}$$

$\psi_i$ is a regularisation term on $x$, and $\lambda_i$ is a weight to balance the importance of regularisation terms and data fidelity. In our deep learning methods, a CNN with leanable parameters is introduced to reconstruct $x$, so the formulation can convert as follows:

$$\hat{x} = \arg\min_x \|M \odot Fx - y\|_2^2 + \lambda\|x - f_{cnn}(x_u|\theta)\|_2^2 \tag{3}$$

here $x_u$ is the zero-filled reconstruction calculated by $x_u = F^H y$ where $F^H$ is inverse Fourier Tranform operator. $f_{cnn}$ represent the forward function of CNN with the parameters $\theta$. In order to generate images like real fully-sampled ones, the optimization of CNN can be written as:

$$\hat{\theta} = \arg\min_\theta \sum_j^N \|x^j - f_{cnn}(F^H y^j|\theta)\|_2^2 \tag{4}$$

with sufficient traning data $\{(x^j, y^j)|j = 1, 2, \cdots, N\}$ and Stochastic Gradient Descent algorithm, CNN can convergence to reasonable state. With fixed CNN, Eq. 3 can be written as:

$$x = f_{dc}(x_{in}, y, M) \quad \text{s.t. } M \odot Fx_{dc} = y \tag{5}$$

where $x_{in} = f_{cnn}(F^H y^j|\theta)$ is the input image reconstructed from CNN. The details of the data consistency layer will be shown in Section 3.5. Furthermore, if the data consistency operation is a determined function to ensure the data fidelity, we can regard it as a part of CNN. And here comes the formulation of our model:

$$\hat{\theta} = \arg\min_\theta \sum_j^N \|x^j - f_{dc}(f_{cnn}(F^H y^j|\theta), y^j, M^j)\|_2^2 \tag{6}$$

## 3.2 Proposed Network

We propose Cascaded Dilated Dense Network with Two-step Data Consistency layer (CDDNwithTDC) for MR image reconstruction. Figure 2 shows an overview of our proposed network, which is composed of a serial of sub-networks. Each sub-network has a De-Aliase Module (DAM) and a Two-step Data Consistency layer (TDC). We use dense block in the DAM and a geometric growth dilation is applied on each dense module for receptive field extension. As MR data is in complex field, we use two channels to represent real part and imaginary part respectively. For example, the input zero-filling image $x_u \in \mathbb{C}^{N_x \times N_y}$ is converted to $x_{input} \in \mathbb{R}^{2 \times N_x \times N_y}$. The details will be described in the following.

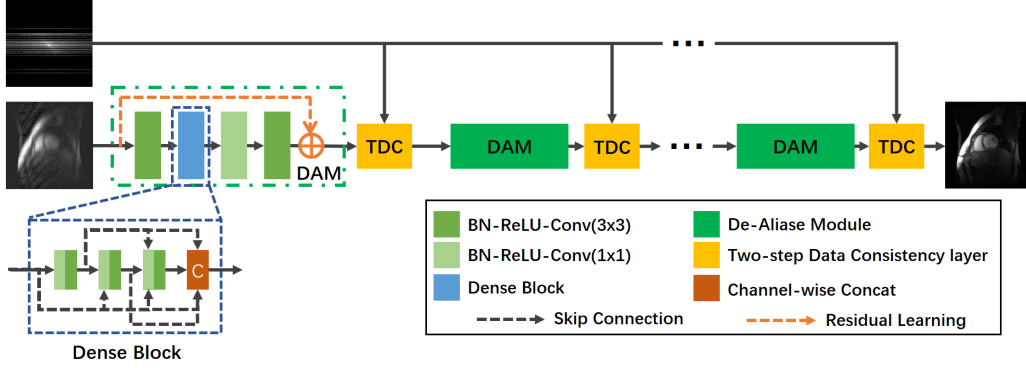

Figure 2: Overview of CDDNwithTDC. We use rectangles with colors to indicate different modules, which is illustrated at the bottom right. A brief illustration of dense block is given at the bottom left.

## 3.3 De-Aliase Module

De-Aliase Module (DAM) is used to generate aliase-free images. The input of the first module is zero-filled MR image while the subsequent modules take the output of former sub-networks as input. The module contains abstraction layer, dense block, transition layer and restore layer. In addition, A global residual connection is applied.

The abstraction layer firstly converts the input image $x_{im} \in \mathbb{R}^{2 \times N_x \times N_y}$ to feature maps $x_{feature} \in \mathbb{R}^{N_f \times N_x \times N_y}$. The forward operation of dense block can be written as $x_j = f([x_0, x_1, \cdots, x_{j-1}])$ where $f$ is convolution operation (called dense layer) and $x_i$ is the output of $i$th layer (specifically, $x_0$ is the input of dense block). The inputs are concatenated in the dimension of channel. $f$ has two parts, the first part is a convolution layer with $1 \times 1$ kernel called bottleneck layer, which reduces the number of feature maps to the original input number (i.e. $N_f$). The second part is a convolution layer with $3 \times 3$ kernel and the outputs have the same number of feature maps. The number ($N_g$) is called as *growth rate*, because the channel of features "grows" layer by layer. All the output of dense layers are concatenated and are fed into a convolution layer with $1 \times 1$ kernel (i.e. transition layer) for halving the number of feature maps. Finally, the restore layer generate output image $x_{out} \in \mathbb{R}^{2 \times N_x \times N_y}$ by a convolution layer with $3 \times 3$ kernel. Notice that every convolution layer is a combination of rectified linear unit activation (ReLU) [4], batch normalization (BN) [8] and convolution neuron.

Dense connection enables intra-block data flow. Such architecture can significantly benefits the performance and robustness. We limit the network parameters on purpose to show that the imporvement is resulted from network architecture rather than simple parameter increment.

## 3.4 Dilated Convolution

With the analogy of the biological term, *receptive field* descripts the area from where artificial neuron abstract information. In other words, it stands for how large portion of image can be seen by a neuron.

Zero-filled MR Images suffer from aliasing artifact. Figure 3 shows an example. With interlaced sampling in *k*-space, the original image occueres in the corrupted image with different offsets in image domain. Notice that additional central phasing-coding lines are fully-sampled as they contains

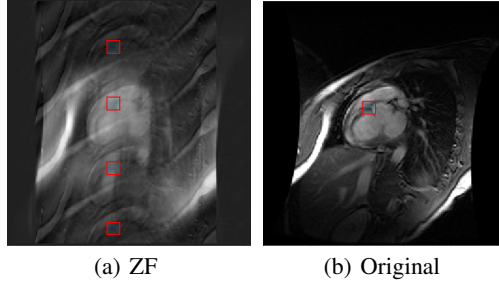

(a) ZF  (b) Original

Figure 3: Aliasing artifact phenomenon. The sampling mask is the same as Figure 1(d). We use red boxes to mark the same low signal area of the original image which can be found several times in the aliased image.

non-sparse low frequency information, so that we can faintly recognize the majority of original image. In order to integrate the scattered many-for-one information, a large receptive field is in need.

We implement the dilated convolution with dense block by appling geometrically increasing (i.e. $1, 2, 4, \cdots$) dilation scale, Figure 4 gives an illustration of so-called Dilated Dense Block. The combination of dilated convolution and dense connection enables Pyramid-like multi-scale feature fusion instead of parallel convolution [15, 35] while keep the depth of network. On the other hand, it successfully expands receptive field without any addition in network parameters.

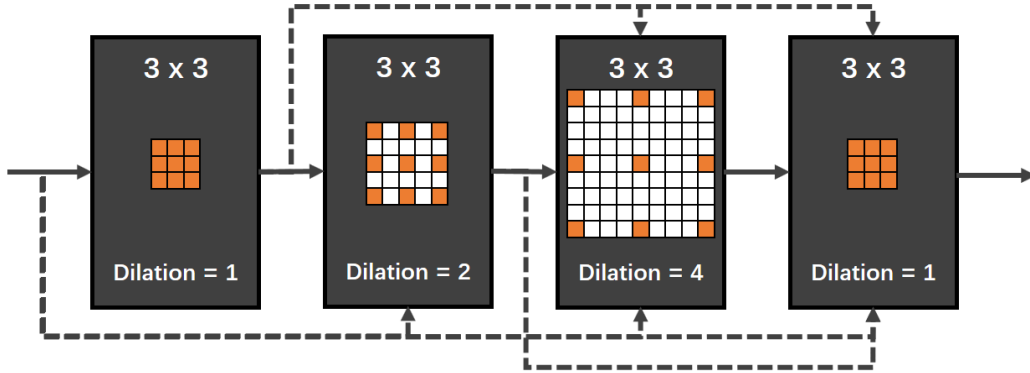

Figure 4: Dilated Dense Block with geometrically increasing dilation scale. The final layer is the restore layer mentioned in Section 3.3, which is 1-dilation convolution to fuse all the former outputs. All the $1 \times 1$ convolution are omitted, not only the bottleneck layers but also the transition layer

### 3.5 Two-step Data Consistency

As mentioned before, MRI acquires data in *k*-space. Data consistency in frequency domain is needed. With fixed parameters $\theta$, Eq.3 has a closed-form solution [18], which can be written as:

$$x_{dc} = F_H \left( (\mathbf{1} - M) \odot F x_{in} + M \odot ( \frac{1}{1+\lambda} F x_{in} + \frac{\lambda}{1+\lambda} y ) \right) \qquad (7)$$

Here $x_{dc}$ is the result image and $\mathbf{1}$ is an all-one matrix. It can be seen as a linear combination taken between $y$ and $Fx$ at the valid position of $M$. Directly replacement is an extreme case with $\lambda = \infty$:

$$x_{dc} = F_H \left( (\mathbf{1} - M) \odot F x_{in} + M \odot y \right) \qquad (8)$$

Unlike traditional image restoration task, the *corrupted* data from sub-sampled MRI is exactly true in the sampled location. During reconstruction, we have to ensure the invariance of the *true* part. Direct replacement can meet the requirement, while it brokes the *self-consistency* of frequency information.

It means the hybird result are unnatural in image-domain. In other word, direct replacement only corrects specific(sampled location) k-space data while leaving others in *outdated* state.

In this paper, we propose a two-step data consistency layer. As shown in Figure 5, we firstly replace corresponding phase-coding lines of generated image $x_{in}$ with the original sampled *k*-space data $y$. Then we convert the result from complex-valued to real-valued format by calculating the modulus $x_m = |x_{dc}|$. In the end, another *k*-space correction is applied on the modulus for data consistency. The two-step data consistency can be formulated as:

$$f_{tdc}(x_{in}, y, M) = F_H\Big((\mathbf{1} - M) \odot F\Big|F_H((\mathbf{1} - M) \odot Fx_{in} + M \odot y)\Big| + M \odot y\Big) \tag{9}$$

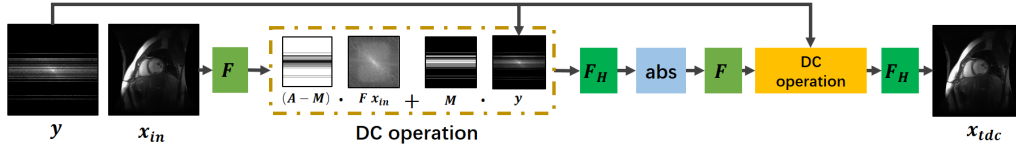

Figure 5: Two-step Data Consistency. The benefits will be evaluated with experiments in Section 4.3.

Empirical experiments prove the effectiveness as shown in Figure 6(a) and further discussion is taken in Section 2 of the Supplementary Material.

## 4 Experiments

### 4.1 Implemetation Details

Experiments are implemented using Pytorch platform on four NVIDIA GeForce GTX 1080Ti with 11GB GPU meomry. Our network is trained with Adam [9] optimizer, initial learning rate is set as 0.0001, the first momentum is 0.9 and the second momentum is 0.999. Weight decay regularization parameter is set as $10^{-7}$. Batch size is 8 and the network is trained for 1000 epochs to ensure convergence.

We cascade 5 sub-networks as default. Each dense block has three BN+ReLU+Conv layers with $1, 2, 4$-dilation, and the growth rate is set as 16. All the convolution layers have 16 feature maps except the last one for mapping from features to two channel images.

Unless otherwise stated, other contrastive networks take the same hyper-parameters. Any notable details will be described in the correponding sub-section.

### 4.2 Dataset

Our dataset, established based on the work of Alexander *et al.* [1], contains 3300 cardiac real-valued MR images from 33 patients. The first 30 patients' data are training set while the last 3 patients are testing set. We use random Cartesian mask with 15% sampling rate like Figure 11(f) as default setting.

### 4.3 Intra-Method Evaluation

In this experiment, we compare the proposed CDDNwithTDC with two variants. One is CDNwithDC, which is implemented without dilated convolution and uses traditional one-step data consistency layer instead. The other network has the Dilated Dense DAM, called CDDNwithDC. We take this experiment to prove the benefits from geometric dilation and two-step data consistency layer. These networks are trained with 30% random Cartesian mask.

Figure 6(a) shows the curve of training MSE loss and Figure 6(b) shows the histogram of testing result, which is taken with two measures, peak signal-to-noise ratio (PSNR) and structural similarity index measure (SSIM) [27]. Dilated convolution can abstract latent information from larger receptive field without parameters increment and TDC can significantly imporve accuracy with negligible computational overhead. Figure 7 gives an example from testing set with 15% sampling rate, indicating that network with TDC result in less reconstruction error.

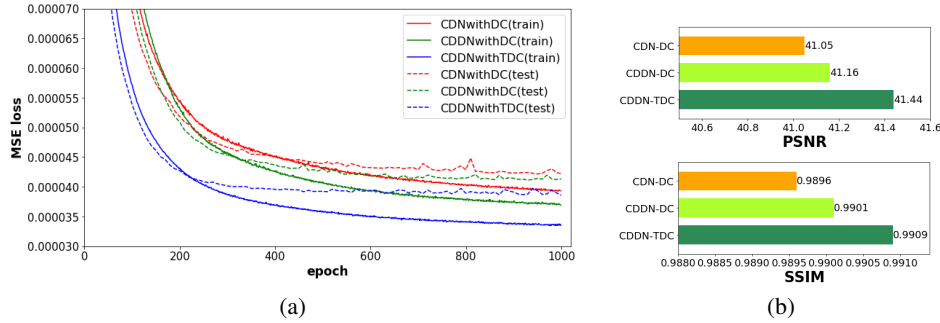

(a)  (b)

Figure 6: Intra-method comparasion. (a) MSE loss. (b) Testing PSNR/SSIM.

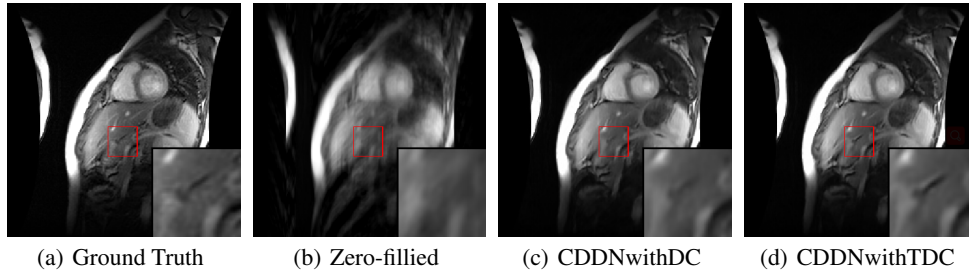

(a) Ground Truth      (b) Zero-fillied      (c) CDDNwithDC      (d) CDDNwithTDC

Figure 7: Benefits from Two-step Data Consistency layer.

## 4.4  Inter-Methods Evaluation

We compare our CDDNwithTDC with deep-learning methods U-Net [7], DC-CNN [21], RDN [22] and conventional methods DLMRI [18] and NLR [3]. DC-CNN is re-implemented according to their paper. With the way of naming in the original paper, we use D5-C5 for 2D reconstruction. As for RDN, we choose the 5B-3D-3R for comparasion, which has the same quantity of network parameters. Figure 8 shows the result on 15% random Cartesian mask. We give a dobozdiagram of general result (Figure 8(a)) and detailed PSNR on every image of 100 testing set (Figure 8(b)) respectively.

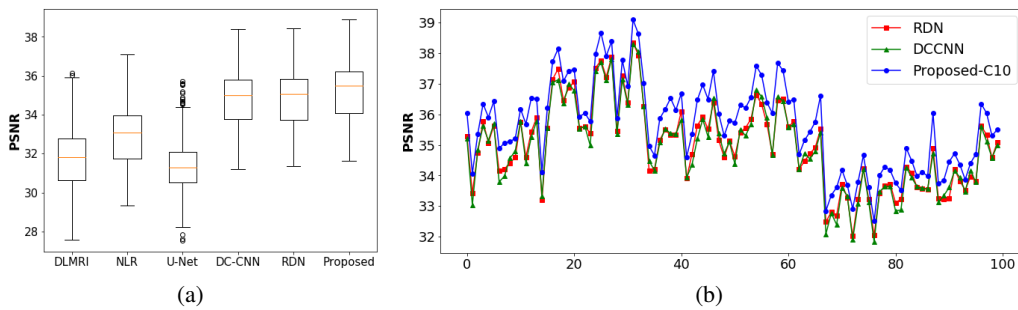

(a)  (b)

Figure 8: The testing result of Inter-Methods Evaluation. As for (b), we show the result from every third image of the 300 testing set, and our method completely exceeds the others.

As our testing set is composed of only three patients, we take 11-fold cross validation experiments in order to alleviate the specificity. Our proposed CDDNwithTDC and DCCNN are re-trained individually 10 additional times for further inter-method comparation. In the $i$th experiment, we take $(i*3-2), (i*3-1), (i*3)$-th patients' data as testing set and the remained 30 patients' data as training set. Figure. 9 shows the result of cross validation, which proves the robustness.

Detailed quantitative comparasion for deep methods is given in Table 1. RDN suffers from long reconstruction time due to its recursive methods. Our proposed method has fewer parameters

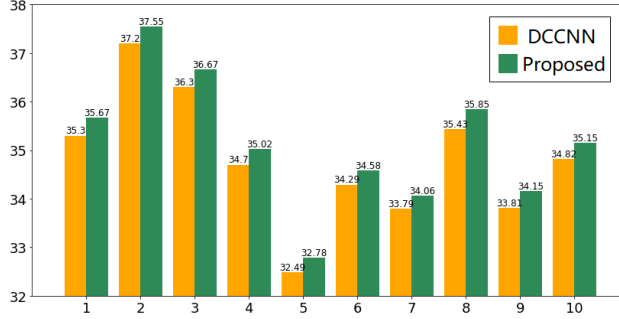

Figure 9: 11-fold Cross Validation.

than DC-CNN/RDN and produces better result. We also cascade 10 sub-networks for our method (Proposed-C10) to reach a comparable number of parameters. An qualitative comparasion is available at Figure 11 as well. We also take experiments on different sampling rate to show the robustness of our method, and the quantitative result is given at Table 2.

Table 1: Comparasion of Deep Methods

| Method | U-Net | DC-CNN | RDN | **Proposed** | **Proposed-C10** |
|---|---|---|---|---|---|
| PSNR | 31.61 | 34.87 | 34.95 | 35.24 | **35.61** |
| Num. of Params. | 1575k | 144k | 144k | **59k** | 119k |
| Train Time($min/epoch$) | 1.1 | 1.2 | 12.0 | 3.0 | 5.8 |
| Test Time($s/frame$) | 0.05 | 0.05 | 0.65 | 0.17 | 0.30 |

## 4.5 Experiment on FastMRI

FastMRI [32] is a dataset of knee MRI. We trained the proposed CDDNwithTDC on part of FastMRI dataset (about 6500 single frame as training set and 700 frames as testing set) to demonstrate the adaptation in different type of MRI. We use the ESC(emulated single-coil) data as ground truth and apply randomly generated mask of 25% sampling rate. Figure. 10(a) shows the loss curves and Figure 10(b) and 10(c) show a qualitative result. It can be seen that our method can reconstructed accuracy details for knee MRI as well as cardiac.

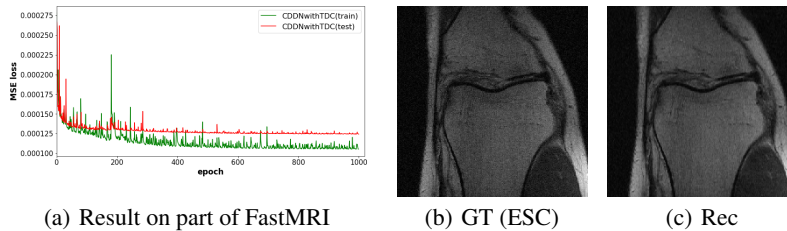

(a) Result on part of FastMRI      (b) GT (ESC)      (c) Rec

Figure 10: Evaluation on FastMRI dataset

## 5 Conclusion

We propose a Cascaded Dilated Dense Network with Two-step Data Consistency layer in MRI reconstruction. Cascading De-Aliase Module based on dense block results in better performance with fewer parameters. Dilated convolution boost the performance of dense blocks. The proposed two-step data consistency layer enhances the result in image domain while keep the complete data consistency in $k$-space. The proposed network achieves state-of-art result and has advantage in the number of network parameters.

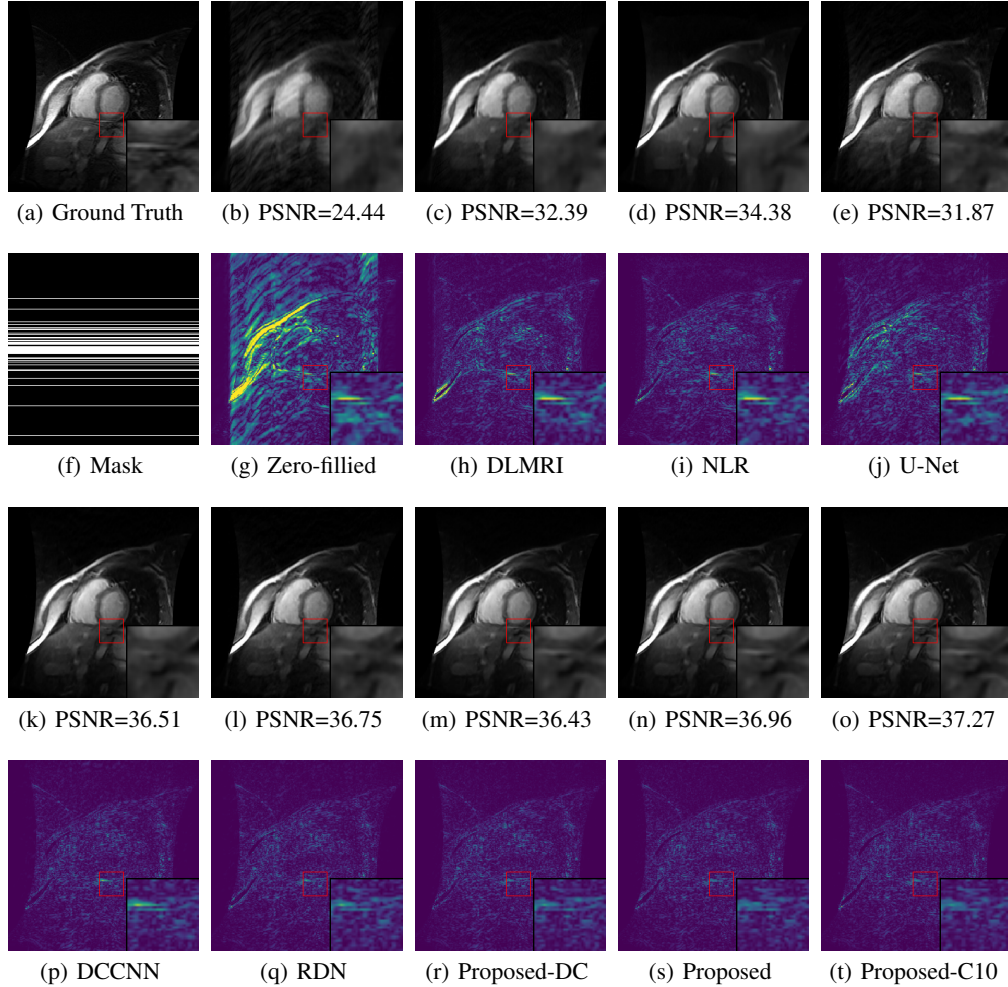

(a) Ground Truth    (b) PSNR=24.44    (c) PSNR=32.39    (d) PSNR=34.38    (e) PSNR=31.87

(f) Mask    (g) Zero-fillied    (h) DLMRI    (i) NLR    (j) U-Net

(k) PSNR=36.51    (l) PSNR=36.75    (m) PSNR=36.43    (n) PSNR=36.96    (o) PSNR=37.27

(p) DCCNN    (q) RDN    (r) Proposed-DC    (s) Proposed    (t) Proposed-C10

Figure 11: Qualitative Comparasion. The 1st, 3rd rows are the reconstructed images and 2nd, 4th rows are the corresponding residual images.

Table 2: PSNR/SSIM Result with Different Sampling Rate

| Method | 2.5% | 5% | 15% | 30% |
|---|---|---|---|---|
| DLMRL | 25.03/0.8179 | 29.46/0.9017 | 31.76/0.9350 | 34.18/0.9548 |
| NLR | 27.30/0.8557 | 31.43/0.9233 | 32.99/0.9461 | 36.66/0.9734 |
| U-Net | 25.96/0.8316 | 29.45/0.8271 | 31.58/0.9312 | 37.24/0.9752 |
| DCCNN | 28.18/0.8872 | 32.24/0.9430 | 34.87/0.9649 | 41.13/0.9900 |
| RDN | 28.29/0.8870 | 31.70/0.9364 | 34.95/0.9665 | 40.54/0.9883 |
| Proposed | 28.43/0.8927 | 32.55/0.9481 | 35.30/0.9689 | 41.66/0.9913 |
| **Proposed-C10** | **28.86/0.9029** | **32.94/0.9526** | **35.60/0.9713** | **42.03/0.9920** |

# 6 Acknowledgements

This work is sponsored in part by the Key Project of the National Natural Science Foundation of China under Grant 61731009, and in part by the National Natural Science Foundation of China under Grant 61871185, and in part by "Chenguang Program" supported by Shanghai Education Development Foundation and Shanghai Municipal Education Commission under Grant 17CG25.

## Footnotes

[2] Our code is released on GitHub:`https://github.com/tinyRattar/CSMRI_0325`

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
