[Supplementary Material · cddn_supplemental.pdf]

# Supplementary Material

## 1 Reduced Dataset

As our original dataset is composed of temporal sequences, the single frames are similar with their adjacent ones. We reduce our dataset by selecting every 1st, 5th, 15th, 20th images of sequences. The reduced dataset has 600 images for training and 60 for testing. We take experiment on both the full dataset and the reduced dataset to show the difference, and the result is given at Figure 1. Unless otherwise state, we use reduced dataset for the experiments in this supplemental material.

Figure 1: The performance will be damaged, while the order of differenct methods keeps in the same.

## 2 Data Consistency

We also try other ways of data consistency. In this section, we will introduce four methods, and we name them from Method I to Method IV.

For Method I, we calculate the modulus $x_m = |x_{in}|$. Then we replace the $k$-space data. Method II is the final edition of the two-step data consistency layer, and the illustration can be found in the paper. Method III adds a $1 \times 1$ convolution layers to fuse the input image (generated by CNN) and the data consistency result of Method I. And Method IV fuse the first DC result and the second one of Method II. An illustration is given at Figure 2

Figure 2: Illustration of Data Consistency Methods.

With cascaded architecture, we propose two varaints that the fusion convolution layers share the same weights. They are called as Method III+ and Method IV+ respectively. Table 1 shows the result. It can be seen that Method II exceeds Method I obviously. Method III gets better result than Method I, while Method II and IV are comparable. Notice that Method IV brings extra parameters to the network. Both the varaints damage the result. This experiment is taken on the reduced dataset with 30% sampling rate with cascaded plain CNN Blocks (five convolution layers).

Table 1: Comparasion of Data Consistency

| Method | Original | I | II | III | IV | III+ | IV+ |
|--------|----------|-------|-------|-------|-------|-------|-------|
| PSNR | 40.77 | 40.78 | 41.26 | 40.95 | 41.27 | 40.74 | 41.10 |

We further make experiment on Method II and Method IV with Dense DAM on the reduced dataset with 15% sampling rate. Method II reachs a result with average PNSR of 34.97 while Method IV is only 34.87. Furthermore, Figure 3 shows that the intermedia images generated by former sub-networks are not de-aliased images but feature maps in Method IV, which is a departure from our purpose. It can be caused by that the parameters of fusion convolution layers are the minor part of the network and they are hard to be optimized. So we take Method II as our determined data consistency layer. Figure 4 shows another experiment which is taken with Method II and Original.

Figure 3: Intermedia result from sub-networks. Row 1 is Method II and the row 2 is Method IV. Column 1 shows the result from first sub-network and column 2 shows the second, and so on.

(a) PSNR=25.89    (b) PSNR=27.12    (c) PSNR=28.87    (d) PSNR=26.11    (e) PSNR=32.26

(f) PSNR=27.53    (g) PSNR=29.65    (h) PSNR=26.09    (i) PSNR=31.30    (j) PSNR=32.90

Figure 4: Residual maps of intermedia result from sub-networks. Row 1 is Original and the row 2 is Method II. The intermedia result might become worse in PSNR as no supervision is applied.

## 3 Benefits of Transition Layer

We make a comparasion on our proposed network with or without transition layer. The transition layer reduces the number of feature maps before the restore layer with $1 \times 1$ convolution. It benefits on the number of network parameters and leads to a better result. Table 2 shows the result. We take experiments on 15% mask with Dense DAM and on 30% with Dilated Dense DAM. Figure 5 gives an illustration of DAM with or without transition layer.

Table 2: PSNR Benefits of Transition Layer

| Sampling Rate | 15%(no Dilation) | 30%(with Dilation) |
|---|---|---|
| Without Transition Layer | 34.58 | 40.87 |
| With Transition Layer | 34.75 | 41.03 |

(a) with transition layer        (b) without transition layer

Figure 5: DAM with/without transition layer.

## 4 Network Architecture

We will introduce the network architecture of the proposed network in detail in Table 3. Here $BN\{N_f\}$ is a batch normalization layer with input channel $N_f$. $Conv\{c_{in}, c_{out}, k \times k, d\}$ is a convolution operation with $c_{in}$ input channels, $c_{out}$ output feature maps and kernel size of $k \times k$, and its dilation scale is $d$.

Table 3: Network Architecture

| | | | | |
|---|---|---|---|---|
| Sub-network 1 | DAM | Abstraction Layer | ReLU | |
| | | | BN{2} | |
| | | | Conv{2, 16, 3 × 3, 1} | |
| | | Dense Block | Bottleneck Layer | ReLU |
| | | | | BN{16} |
| | | | | Conv{16, 16, 1 × 1, 1} |
| | | | Convolution Layer | ReLU |
| | | | | BN{16} |
| | | | | Conv{16, 16, 3 × 3, 1} |
| | | | Bottleneck Layer | ReLU |
| | | | | BN{32} |
| | | | | Conv{32, 16, 1 × 1, 1} |
| | | | Convolution Layer | ReLU |
| | | | | BN{16} |
| | | | | Conv{16, 16, 3 × 3, 2} |
| | | | Bottleneck Layer | ReLU |
| | | | | BN{48} |
| | | | | Conv{48, 16, 1 × 1, 1} |
| | | | Convolution Layer | ReLU |
| | | | | BN{16} |
| | | | | Conv{16, 16, 3 × 3, 4} |
| | | Transition Layer | ReLU | |
| | | | BN{64} | |
| | | | Conv{64, 16, 1 × 1, 1} | |
| | | Restore Layer | ReLU | |
| | | | BN{16} | |
| | | | Conv{16, 2, 3 × 3, 1} | |
| | Two-step Data Consistency Layer | | | |
| Sub-network 2 | . . . | | | |
| Sub-network 3 | . . . | | | |
| Sub-network 4 | . . . | | | |
| Sub-network 5 | . . . | | | |

## 5 Network Parameters

Our methods can achieve better results with fewer parameters. Beside the proposed network (cascaded 5 sub-networks), we also use Proposed-C10 (cascaded 10 sub-networks) as an instance with comparable network parameters with DCCNN. DCCNN is reimplemented with different numbers of intermedia feature maps. Here are two variants, one is DCCNN-f16 with 16 feature maps and the other is DCCNN-f64. Notice that the original DCCNN uses 32 feature maps. U-Net is taken as a benchmark.

We show the relationship of network parameters and result PSNR in Figure 6. The experiment is taken with 30% sampling rate. It can be seen that our methods exceed others in both performance and network parameters.

Figure 6: The comparasion of network parameters and performance.

## 6 Effect of Cascading Iterations

In this section, we investigate how the cascading iteration affects performance. We increasingly cascading sub-networks from 1 to 10 instances. The training dataset is the full set with 15% sampling rate. Although the effect of additional blocks decreases rapidly, it can stably imporve results. Such cascading leads to linealy increasing number of network parameters, so we choose 5 cascaded sub-networks as our determined architecture. We give the curve of result PSNR and cascading iterations in the Figure 7.

Figure 7: The relationship of performance and cascading iterations.

## 7 Sampling Rate

Figure 8 gives result of the proposed network trained with different sampling rate, which has a positive relationship with reconstruction result. Our method can handle a large range of sampling rate, even aggressive one (5%).

|  |  |  |  |
|---|---|---|---|
| (a) 5% | (b) PSNR=25.36 | (c) PSNR=29.70 | (d) |
| (e) 10% | (f) PSNR=26.99 | (g) PSNR=32.70 | (h) |
| (i) 15% | (j) PSNR=28.30 | (k) PSNR=35.95 | (l) |
| (m) 30% | (n) PSNR=33.00 | (o) PSNR=42.91 | (p) |

Figure 8: Reconstruction with Different Sampling Rate. Column 2 are zero-filled reconstruction and column 3 are our results. Residual maps of our results is given at column 4.

## 8 Overall Visual Result

We show the collection of the reduced testing dataset in Figure 9. Our proposed method can accurately reconstruct the de-aliased MR images. The network is trained on reduced dataset with 15% sampling rate.

(a) Label

(b) Reconstructed Result

(c) Residual Map

Figure 9: Collection of Reduced Testing Set

# 9 More Results

This section is a supplementary of Section 4.4 in the paper. We will show more results to prove the advantages of the proposed method. More visual result will be given in the following.

Figure 10 shows an additional qualitative evaluation with deep methods. The results come from the 212th image of the full test dataset and the networks are trained with 15% sampling rate. Although

69  the results in PSNR are close among the last three methods, our method can preserve more details
70  during the reconstrcution as shown in the zoom-in images.

(a) GT    (b) PSNR=26.42 (c) PSNR=29.97 (d) PSNR=32.37 (e) PSNR=32.29 (f) PSNR=32.53

(g) Mask    (h) ZF    (i) U-Net    (j) DCCNN    (k) RDN    (l) Proposed

Figure 10: Qualitative Result From 212th Image. The first row are the reconstructed results while the second row are the residual maps.

71  Figure 11 gives the results from 19th image. The performance of RDN can be unstable as DCCNN
72  generate better images than RDN in Figure 10, while the proposed network has robustness and can
73  generate better result in visual.

(a) GT    (b) PSNR=24.51 (c) PSNR=31.77 (d) PSNR=35.31 (e) PSNR=35.60 (f) PSNR=35.96

(g) Mask    (h) ZF    (i) U-Net    (j) DCCNN    (k) RDN    (l) Proposed

Figure 11: Qualitative Result From 19th Image