[Reviews · NeurIPS 2019]

Reviewer 1



Originality: CDDNwithTDC is new and novel: its cascading dense blocks with dilated convolution and its two-step data consistency layer. The author adequately cites relevant works. Quality/Clarity: This is a complete piece of work. The authors are honest about the proposed solution's strengths and weaknesses. The solution may take longer to train and test than some of the prior arts. Nonetheless, the solution uses much fewer number of parameters while setting the new state of the art as far as its performance (PSNR and SSIM) is concerned. The authors also provide an example image with the lesion region zoomed in so that the reader can qualitatively compare the effect of the extra data consistency step in the proposed model. The experiments are also thorough. It showed the effect of the sampling rate and the number of parameter on the model performance (PSNR). The mathematical formulation is sound and clear as well. Overall, it was pleasant reading and following the paper. Significance: The proposed solution and the results reported in the paper are very significant from both algorithmic and application perspectives.

Reviewer 2



The authors have done a commendable job at developing a model to reconstruct MRI from k-space and evaluating it quantitatively. I'm raising the following possible questions/concerns: - Since the test set is small (3 patients), I would have liked to see k-fold cross validation. - A fair test to compare the reconstructed image to the ground truth could be radiologists' interpretation of them. Quantitative and anecdotal qualitative assessment is great, but it is also important to know if the radiologists' interpretation would have been consistent between the reconstructed and ground truth images. - Have the authors looked into using the FastMRI dataset? I do not have a clinical background, but I wonder if the FastMRI dataset could have been used as a second dataset for reproducibility and generalization of the proposed approach.

Reviewer 3



I found this paper hard to read. Although I am familiar with both the physics of MRI and deep learning (and have worked at the intersection of the two), it was difficult to follow the exact details of the optimization objective, the data consistency term, etc. More generally, this paper seems to be only a marginal advance over other methods. It seems to fall into the category of "Here is a new DNN architecture with features tuned to a specific problem, and some experiments showing that it works." There is no theoretical justification for the choices made, although they seem reasonable given the context of the problem statement -- however, the final experiments show that the method performs only slightly better than competing DNN-based reconstruction methods, and given the error bars show, it is not clear that the results are statistically significant. Given the readability issues; the ad hoc nature of the architecture; and the less-than-stellar results, I do not believe that this paper crosses the bar for publication in NeurIPS.

[Author Response · NeurIPS 2019]

**Thank you for your concerns. We hope this response can address all your questions. Best wish :)**

**For Reviewer #1:** Thank you for your great approval. For the first question, Figure 8(b) in the original paper shows that the proposed method outperforms others. We have also checked more randomly selected samples. Honestly, some samples have similar results visually, especially the ones with fewer details. Like Figure 9(q) and 9(s) in the paper, the difference of error maps is not significant while the PSNR is 0.21db better.The former has lower intensity visually in the top-right area but it leaves more aliased artifact in the top-middle area. However, both areas are not important for clinical diagnosis and our methods has more distinct edge in the zoom-in area. As for lesion images, our methods can restore details with enough sampling rate (like 15%). While with aggressive sampling rate, the details might be hard to reconstruct because of the lack of high frequecy phasing lines as shown in the Figure 8 of the supplementary material. Anyway, the low frequency part are de-ailiased well. Up to now, we haven't noticed any poor result in our dataset yet. The checking process will be continued, and the trained weights will be released with our codes if the paper is accepted.

**For Reviewer #2:** Thank you for your suggestions. We take 11-fold cross validation experiments on our dataset. The result of PSNR is shown in Figure (a) and it demonstrates the robustness. As MRI reconstruction serves clinic diagnosis, radiologists' interpretation is indeedly a more reasonable standard. A formal surver for radiogists should be taken in the future. Acutally, we showed our result to some clincians and won their recognition. Our result have rather good quality and keep consistency with groud truth in distinguishing the edge of tissue is distinct or not, which is a major evidence for diagnose of certain diseases. We hope to conduct further experiments on the FastMRI dataset but it will take a long time for training. Because of our insufficiency of GPU devices and the limitaion of time, what we can show is the result trained with FastMRI eval dataset. The loss curves are shown at Figure (b), we believe further training and fine-tuning on hyper-parameters will promote the convergence. An qualitative result is shown in Figure (c) and (d). It can be seen that our method can reconstructed accuracy details for FastMRI dataset. The training code for FastMRI will be available as well if the paper is accepted and the training process will be continued.

(a) 11-fold Cross Validation  (b) Result on Reduced FastMRI  (c) GT (esc)  (d) Rec

**For Reviewer #3:** I'm sorry for your confusion. A revision will be taken on the paper and we're trying to provide more details and emphasize key points here. In this paper, we propose a novel DNN architecture for MRI reconstruction. Our main contributions are (1) Cascaded dense blocks, (2) combination of dilation convolution and dense connection, (3) novel Two-Step Data Consistency layer. Cascading technology is proved to be effective in our experiments and other related works as it alleviates the difficulty of residual learning. Discussion of the benefits can be found in the Section 6 of the Supplementary Material. As mentionded in Section 3.4 of the paper, dilation convolution can expand receptive field for the demand of de-aliasing task. With dense connection, the feature maps from different receptive field sizes can be fused together for reconstruction. Experiment in Section 4.3 shows the necessity. All the designs of De-Aliase Module are proposed base on above analysis and are proved to be effective by extensive experiments in the paper.

As for data consistency term, the first thing to confirm is that the raw MRI data is k-space (frequency-domain) data. Unlike traditional image restoration task, the *corrupted* data is exactly true in the sampled location. During reconstruction, we have to ensure the invariance of the *true* part (i.e. the first term in Eq.2 of the paper). Direct replacement can meet the requirement, while it brokes the *self-consistency* of frequency information. It means the hybird result are unnatural in image-domain. In other word, direct replacement only corrects specific(sampled location) k-space data while leaving others in *outdated* state. As stated in Section 3.2 of the paper, we use complex-value structure to preserve information. The complex part of intermedia result can benefits the method with its latent information. However, only real-valued image is accepted in Human Visual System. In our data consistency unit, we apply modulus calculation on the firstly corrected data to imporve image-domain reasonablity. Finally, another direct replacement is taken for ensuring invariance as we have no reason to abandon *exactly true* data. Empirical experiments prove the necessity. Section 3.5 states our TDC module, Figure 7 illustrates the benefits and Section 2 of the Supplementary Material takes further discussion. We hope these explanations can help you follow the exact details.

Our advantage is reasonable in the present state of MRI reconstruction research. As shown in the Table 2 of the paper, our method(Proposed-C10) outperforms the secondary method by about 0.7db in PSNR. The minimun is 0.68db while the maximin is 0.80db. Accroding to recently published articles of MRI reconstruction, even if the advantage is hardly "very significant", we thought it can be "pretty good". Figure 9(t) also shows our advantage in the detail restoration by comparing with 9(p) and 9(q). More validation experiments in **"For Reviewer #2"** show the robustness and superiority.

[Meta-Review · NeurIPS 2019]

The scores on this paper is mixed with two reviewers in clear favour of acceptance and one reviewer with a strong reject. The latter reviewer unfortunately did not contribute to the discussion. After looking at the paper, reviews, rebuttal and discussion, I find the paper is thorough and proposes a good solution to an important and therefore recommend acceptance despite the average score being below the usual cut. The median (7) is clearly within acceptance range.